# Serum Metabolomic Analysis of Synchronous Estrus in Yaks Based on UPLC-Q-TOF MS Technology

**DOI:** 10.3390/ani14101399

**Published:** 2024-05-07

**Authors:** Fen Feng, Chun Huang, Dunzhu Luosang, Xiaoming Ma, Yongfu La, Xiaoyun Wu, Xian Guo, Zhandui Pingcuo, Chunnian Liang

**Affiliations:** 1Key Laboratory of Yak Breeding of Gansu Province, Lanzhou Institute of Husbandry and Pharmaceutical Sciences of Chinese Academy of Agricultural Sciences, Lanzhou 730050, China; feng990111@163.com (F.F.); johnchun825@163.com (C.H.); maxiaoming@caas.cn (X.M.); layongfu@caas.cn (Y.L.); wuxiaoyun@caas.cn (X.W.); guoxian@caas.cn (X.G.); 2Key Laboratory of Animal Genetics and Breeding on Tibetan Plateau, Ministry of Agriculture and Rural Affairs, Lanzhou 730050, China; 3Plateau Agricultural Science and Technology Innovation Center, Lasa 850004, China; 4Institute of Animal Husbandry and Veterinary Medicine, Tibet Academy of Agriculture and Animal Husbandry Sciences, Lasa 850004, China; sontsa76@163.com

**Keywords:** yak, synchronous estrus, serum, UPLC-Q-TOF MS, metabolism

## Abstract

**Simple Summary:**

The synchronization process of yak estrus plays a crucial role in enhancing the reproductive success rate of yaks and safeguarding the continuation of the species. Metabolomics, a branch of bio-chemistry, primarily focuses on the quantitative analysis of all metabolites within an organism, aiming to identify the relative relationships between metabolites and physiological changes. It serves as a valuable tool for delving into the molecular mechanisms and responses of metabolic pathways under various perturbations. In this study, metabolomics methods were utilized to analyze blood samples collected from non-estrus yaks as well as those that underwent hormonally induced synchronized estrus to identify differential metabolites. The research results provide a theoretical basis for the optimization and application of synchronized estrus technology in yaks, promoting the healthy development of the yak industry.

**Abstract:**

The yak is a unique species of livestock found in the Qinghai-Tibet Plateau and its surrounding areas. Due to factors such as late sexual maturity and a low rate of estrus, its reproductive efficiency is relatively low. The process of estrus synchronization in yaks plays a crucial role in enhancing their reproductive success and ensuring the continuation of their species. In order to clarify the characteristics of the serum metabolites of yak estrus synchronization, the yaks with inactive ovaries were compared with the estrus synchronization yaks. In this study, yaks were divided into the inactive ovaries group (IO), gonarelin-induced yak estrus group (GnRH), and chloprostenol sodium-induced yak estrus group (PGF). After the completion of the estrus synchronization treatment, blood samples were collected from the jugular veins of the non-estrus yaks in the control group and the yaks with obvious estrus characteristics in the GnRH and PGF groups. Metabolites were detected by ultra-high performance liquid chromatography-mass spectrometry, and differential metabolites were screened by multivariate statistical analysis. The results showed that a total of 70 significant differential metabolites were screened and identified in the GnRH vs. IO group, and 77 significant differential metabolites were screened and identified in the PGF vs. IO group. Compared with non-estrus yaks, 36 common significant differential metabolites were screened out after the induction of yak estrus by gonarelin (GnRH) and cloprostenol sodium (PGF), which were significantly enriched in signaling pathways such as the beta oxidation of very long chain fatty acids, bile acid biosynthesis, oxidation of branched chain fatty acids, steroidogenesis, steroid biosynthesis, and arginine and proline metabolism. This study analyzed the effects of gonadotropin releasing hormone (GnRH) and prostaglandin F (PGF) on the reproductive performance of yaks treated with estrus synchronization, which provides a theoretical basis for the optimization and application of yak estrus synchronization technology and promotes the healthy development of the yak industry.

## 1. Introduction

The yak, also known as the “boat of the plateau”, is a seasonal breeding mammal mainly distributed in the Qinghai-Tibet Plateau area [1]. Yaks, due to their ability to adapt to high-altitude, cold, nutrient deficient, and hypoxic environments, provide the necessary basic resources (such as meat, milk, transportation, and fuel manure) for local herders. They have always been considered the most important means of production and livelihood for local herders, and are an indispensable and important breed of livestock [2]. However, due to the harsh natural conditions, most yaks suffer from various reproductive problems compared to ordinary cattle living in plain areas such as late sexual maturity, long calving intervals, and low estrus rates, leading to low reproductive efficiency and severely restricting yak production [3,4]. Therefore, with the development of embryonic biotechnology, techniques such as sympathetic estrus, superovulation, and embryo transfer are crucial for accelerating breeding progress, improving yak fertility, and protecting yak germplasm resources.

The synchronous estrus technique is based on the regulatory mechanism of reproductive hormones on follicle development during the estrus cycle, using exogenous reproductive hormones to treat female animals, extending or shortening the estrus period of some female animals in the herd to synchronize a group of female animals in a relatively short time, reduce the workload of estrus identification, and improve the estrus detection rate, estrus rate, and reproductive rate of female animals [5,6]. Gonadotropins and prostaglandins are commonly used hormones for estrus synchronization in female animals [7,8]. Research reports have indicated that the use of GnRH and PGF2α treatment is a practical method for improving the efficiency and accuracy of synchronized estrus in beef cattle [9]. Meanwhile, GnRH and PGF2α play important roles in estrus and ovulation in yaks, and monotherapy with GnRH or PGF2α + GnRH can successfully induce estrus in yaks that had calved in previous years, but did not calve that year [4,10].

Metabolomics, a high-throughput technology rooted in nuclear magnetic resonance (NMR) spectroscopy and mass spectrometry, is dedicated to the comprehensive analysis of small molecules. This approach encompasses both targeted and untargeted strategies, rendering it a formidable tool for dissecting metabolic alterations across diverse biological contexts [11,12]. Organisms experience variations in their metabolites due to hormonal induction, physiological or pathological stimuli, and environmental changes. Organisms are subject to hormone induction, physiological or pathological stimulation, and environmental changes, which lead to changes in their metabolites [13,14,15]. Therefore, metabolomics serves as a valuable tool for identifying novel and potential metabolite markers and delving into the molecular mechanisms and responses of metabolic pathways in response to various perturbations. UHPLC-Q-TOF/MS is the most commonly used method in non-targeted metabolomics. The methodology leverages two distinct chromatographic columns to conduct a thorough analysis of small molecule metabolites, capturing data in both positive and negative ion modes with high resolution, rapid throughput, and exceptional sensitivity [16,17]. However, the differences in the functional metabolites of yak serum treated with different drugs have not been well-characterized. Therefore, this study used UHPLC-Q-TOF/MS technology to investigate the differential metabolism in serum samples of yak estrus treated with different drugs, in order to determine which metabolite is most suitable for synchronous estrus in yaks.

## 2. Materials and Methods

Guided by the Animal Ethics Procedures and Guidelines of the People’s Republic of China, all yak handling strictly adhered to good animal practices. The study was approved by the Animal Administration and Ethics Committee of Lanzhou Institute of Husbandry and Pharmaceutical Sciences of the Chinese Academy of Agricultural Sciences (Lanzhou, China) (Permit No. 2020-015).

### 2.1. Experimental Animals and Sample Collection

The study selected 45 female yaks from Gansu Province’s Gannan Tibetan Autonomous Prefecture in August 2020, ensuring that they were in good physical condition, similar in body type, had adequate nutritional status, and were free from reproductive system diseases to serve as the experimental subjects. Before the onset of estrus in female yaks during the estrus season, 15 yaks were randomly selected for intramuscular injection of sodium cloprostenol (Ningbo Sansheng Biotechnology Co. Ltd., Ningbo, China) as the experimental NE group, and 15 yaks were randomly selected for intramuscular injection of gonarelin (Ningbo Sansheng Biotechnology Co. Ltd.) as the experimental GnRH group; the remaining 15 yaks with inactive ovaries were not treated as the IO group (control group). Subsequently, observing the estrus of the recipient yak after completing the synchronous estrus treatment, the recipient yaks exhibited exposed vulva and stably stood to accept the mounting behavior, indicating estrus. Ten yaks with obvious estrus characteristics were selected from each of the GnRH and PGF groups, and 10 non-estrus yaks were selected from the control group. After the completion of the estrus synchronization treatment, blood samples were collected from the jugular veins of the non-estrus yaks in the control group and the yaks with obvious estrus characteristics in the GnRH and PGF groups. A total of 20 mL of blood was collected from each yak, centrifuged at 4000× *g* for 10 min, serum samples taken, and then stored them at −80 °C.

### 2.2. Extraction of Metabolites

The mixture was sonicated on ice for 60 min and allowed to stand for 1 h. At 4 °C, 16,000× *g* centrifuged for 20 min, the supernatant was transferred to the sample tube, and the supernatant was concentrated and dried in a high-speed vacuum concentration centrifuge. A total of 100 μL acetonitrile–water solution (1:1, *v*/*v*) was used to dissolve the dried extract, centrifuged at 16,000× *g* at 4 °C for 15 min, and then detected. The resulting mixture was sonicated in an ice bath for 60 min and then incubated at −20 °C for 1 h. Subsequently, it was centrifuged at 16,000× *g* and 4 °C for 20 min. The supernatant was carefully inhaled into a new sample tube and concentrated and dried in a high-speed vacuum centrifuge. In order to re-suspend the dried extract, 100 μL of acetonitrile–water solution (1:1, *v*/*v*) was added. The solution was centrifuged at 16,000× *g* at 4 °C for 15 min and prepared for analysis.

### 2.3. Extraction of Metabolites UHPLC-Q-TOF/MS Analysis

An Agilent 1290 Infinity LC ultra-high performance liquid chromatography (UHPLC) system equipped with a hydrophilic interaction liquid chromatography (HILIC) column was utilized for the analyses. The mobile phase flow rate was maintained at 0.3 mL/min, with the column temperature stabilized at 25 °C. Sample injections were performed using 5 μL aliquots. The mobile phase composition comprised solvent A, a mixture of water, 25 mM ammonium acetate, and 25 mM ammonia, and solvent B, acetonitrile. Gradient elution procedures: 0–0.5 min, 95% B; 0.5–7 min, 95–65% B; 7–9 min, 65%–40% B; 9–10 min, 40% B; 10–11.1 min, 40–95% B; 11.1–16 min, 95% B. The positive and negative ion modes were detected by point spray ionization (ESI), and the mass spectrometry was analyzed by a Triple-TOF 5600 mass spectrometer (AB SCIEX) after UPLC separation. The ESI source conditions were as follows. Ion Source Gas1 (Gas1)/Ion Source Gas2 (Gas2), 60 psi; source temperature, 600 °C; IonSapary Voltage Floating (ISVF) ± 5500 V (positive and negative two modes); TOF MS scan *m*/*z* range: 60–1200 Da, production scan *m*/*z* range: 25–1200 Da, TOF MS scan accumulation time 0.15 s/spectra, product ion scan accumulation time 0.03 s/spectra. The secondary mass spectrum was generated through information-dependent acquisition (IDA), employing high sensitivity mode. The deconvoluting potential (DP) was set to ±60 V for both positive and negative ionization modes, while the collision energy was fixed at 30 eV. The IDA parameters were configured so that ions within a 4 Da range were excluded from subsequent analysis, and the number of candidate ions to be monitored per cycle was set at 6 [18,19].

### 2.4. Data Pre-Processing and Multivariate Statistical Analysis

Peak alignment, retention time adjustment, and peak area extraction were accomplished using the XCMS module within the Mass Spectrometry-Data Independent AnaLysis (MS-DIAL) (ver. 2.62) software suite. The structural elucidation of metabolites was conducted based on high-accuracy mass measurements (<25 ppm tolerance) and subsequent comparison with reference spectra. This process involved matching against comprehensive databases such as the Human Metabolome Database (HMDB), MassBank, and a personally curated metabolite library, ensuring a robust identification methodology. The 50% ion peak was adopted as the benchmark, and missing values within the group were subsequently removed. Following normalization, the data underwent Pareto scaling utilizing SIMCA-P14.1 software (Umetrics, Umea, Sweden). Subsequent multivariate analyses including principal component analysis (PCA), partial least squares-discriminant analysis (PLS-DA), and orthogonal partial least squares-discriminant analysis (OPLS-DA) were then conducted.

### 2.5. Metabolite Identification and Pathway Analysis

Metabolites exhibiting significant alterations were identified using a fold change (FC) threshold of >2 or <0.5, coupled with a *p* < 0.05. Additionally, metabolites with a variable importance in projection (VIP) score exceeding 1, along with a t-test *p* < 0.05 derived from the OPLS-DA model, were considered significantly distinct. These metabolites were then subjected to hierarchical clustering analysis and Kyoto Encyclopedia of Genes and Genomes (KEGG) metabolic pathway analysis to elucidate their roles and interactions within biological systems. This analytical approach facilitates the interpretation of complex biological data, enhancing our understanding of underlying metabolic processes and their regulation.

## 3. Results

### 3.1. Analytical Evaluation of System Stability

#### 3.1.1. Comparison of Total Ion Diagrams of QC Samples

Mass spectral TIC plots for quality control (QC) samples in both positive and negative ion detection modes were subjected to comparative overlapping analysis. The findings revealed that the chromatographic peak response intensities and retention times were consistently overlapping, suggesting minimal instrument-induced variability throughout the experiment (Figure 1). Consequently, the data integrity was deemed trustworthy, affirming the robustness of the analytical methods employed.

#### 3.1.2. Principal Component Analysis (PCA) of Total Samples

Utilizing MS-DIAL software, ion peaks corresponding to metabolites were meticulously extracted from the experimental samples. Following Pareto scaling, these peaks were subjected to principal component analysis (PCA), and the PCA model was validated through seven cycles of seven-fold cross-validation. As illustrated in Figure 2, the tight clustering of QC samples underscores the high experimental repeatability and the robust stability of the instrumental analytical system. This collective stability and reliability of the experimental data affirm that the metabolic spectrum variations observed truly reflect inherent biological disparities among the individual samples.

### 3.2. Multivariate Statistical Analysis

Contrary to PCA, PLS-DA serves as a supervised discriminant analysis technique that establishes a predictive model linking metabolite expression levels to sample classifications via partial least squares regression. The PLS-DA model scores are depicted in Figure 3, derived after a rigorous process of seven-fold cross-validation. The resulting model evaluation metrics, R2Y and Q2, were calculated the model data for each group. For the GnRH vs. IO group, the values were R2Y = 0.991 and Q2 = 0.688, and for the PGF vs. IO group, R2Y = 0.996 and Q2 = 0.767. With both R2Y and Q2 exceeding 0.5, this signifies that the models possess strong explanatory power and predictive capabilities, rendering them reliable and stable. Building upon the foundation of PLS-DA, the OPLS-DA method was employed to refine the data, filtering extraneous noise unrelated to classifying information to enhance the model’s explanatory and predictive power, and directly elucidate intergroup variations. Moreover, to ascertain whether the established model exhibited signs of ‘overfitting’, the OPLS-DA model underwent a permutation test. As delineated in Figure 4, the OPLS-DA model effectively segregated samples into distinct groups, and when coupled with the permutation test, the Q2 intercepts for both the GnRH vs. IO and PGF vs. IO groups were less than 0.05, confirming the absence of overfitting within the constructed OPLS-DA model. Consequently, the VIP score—a metric gauging the impact of metabolite expression patterns on sample categorization and the explanatory capacity of metabolites—could be computed, facilitating the identification and screening of differential metabolite markers.

### 3.3. Screening of Differential Metabolites

The preprocessed data were analyzed using the *t*-test and analysis of variance multiplicity, and volcano plots were plotted with FC and the *p* value to screen for differential metabolites (FC > 2 or FC < 0.5 with *p* < 0.05 as the screening criteria). Figure 5A depicts the volcano plot for the GnRH vs. IO group, while Figure 5B presents the volcano plot for the PGF vs. IO group. In these figures, red dots represent upregulated metabolites, and green dots represent downregulated metabolites, which are the differential metabolites identified through univariate statistical analysis. The results showed that all metabolites were normally distributed, indicating that the data could be used for further screening and identification. VIP values derived from the OPLS-DA model served to quantify the impact of each metabolite’s expression pattern on the classification of sample groups and the metabolites’ explanatory power, thereby uncovering biologically relevant differential metabolites. Metabolites exhibiting both a VIP > 1 in multivariate statistical analysis and a *p* < 0.05 in univariate statistical analysis were identified as having significant differences. These findings are outlined in Appendix A. The analysis revealed the detection of 70 significantly varying metabolites between the GnRH and IO groups; conversely, 77 such metabolites were distinguished in the comparison between the PGF and IO groups.

### 3.4. Characterization and Function of Metabolic Pathways in Paired Comparison of GnRH vs. IO

To illustrate the expression profiles of differential metabolites across various samples, hierarchical clustering was conducted on each group by utilizing the qualitative importance of differential metabolite expression levels. Metabolites congregating within the same cluster display analogous expression patterns and might occupy proximate reaction stages within the metabolic pathway. As shown in Figure 6A, the GnRH vs. IO group data could be well-clustered together, indicating that there was no significant difference within the experimental sample group and that similar metabolites were clustered together. Significantly altered metabolites identified from comparative analyses were subjected to KEGG ID mapping and uploaded to the KEGG database for comprehensive pathway enrichment investigations (Figure 6B). The results showed that significantly different metabolites mainly participated in the following metabolic pathways: bile acid biosynthesis, beta oxidation of very long chain fatty acids, phenylacetate metabolism, betaine metabolism, oxidation of branched chain fatty acids, phospholipid biosynthesis, gluconeogenesis, retinol metabolism, methionine metabolism, steroidogenesis, pyruvate metabolism, glycine and serine metabolism, tryptophan metabolism, and so on.

### 3.5. Identification and Functional Analysis of Significantly Different Metabolites in the Pairwise Comparison of PGF vs. IO

Based on qualitative significant differences in metabolic expression levels, hierarchical clustering was performed on the PGF vs. IO group samples (Figure 7A). The results showed that the PGF vs. IO group data could be well-clustered in the instrument, indicating no significant differences within the experimental sample group. KEGG pathway enrichment analysis was performed on significantly different metabolites. As shown in Figure 7B, significantly different metabolites are involved in 25 different metabolic pathways including bile acid biosynthesis, beta oxidation of very long chain fatty acids, homocysteine degradation, methionine metabolism, D-arginine and D-ornithine metabolism, steroidogenesis, phosphatidylethanolamine biosynthesis, steroid biosynthesis, phosphatidylcholine biosynthesis, pyrimidine metabolism, glutathione metabolism, threonine and 2-oxbutanoate degradation, betaine metabolism, transfer of acetyl groups into mitochondria, glycerolipid metabolism, oxidation of branched chain fatty acids, plasmalogen synthesis, selenoamino acid metabolism, urea cycle, mitochondrial beta-oxidation of long chain saturated fatty acids, phospholipid biosynthesis, citric acid cycle, fatty acid elongation in mitochondria, fatty acid biosynthesis, and propanoate metabolism.

### 3.6. Identification and Functional Analysis of Metabolites with Common Significant Differences between GnRH vs. IO and PGF vs. IO

It can be seen from Figure 8A that there were 36 different metabolites in common in GnRH vs. IO and PGF vs. IO, which are as follows: hydroxyproline, acetylcarnitine, auranticin A, but-3-enylglucosinolate, butyryl carnitine, caffeic acid, capsanthone, cholesterol, cholic acid, dehydrorotenone, and so on. KEGG analysis of common differential metabolites showed that there were six differential metabolic pathways (Figure 8B), which were: beta oxidation of very long chain fatty acids, bile acid biosynthesis, oxidation of branched chain fatty acids, steroidogenesis, steroid biosynthesis, and arginine and proline metabolism.

## 4. Discussion

In recent years, metabolomics technology has garnered widespread application across numerous disciplines, emerging as a pivotal tool for dissecting disease etiologies and identifying potential biomarkers. Metabolomics has been validated as a robust and insightful methodology for pattern recognition analyses within diverse biological contexts, particularly through liquid chromatography/mass spectrometry (LC/MS)-based approaches, which allow for the comprehensive and quantitative assessment of myriad metabolic biomarkers in biological specimens [20,21,22]. This study used UHPLC-Q-TOF MS technology to analyze the serum metabolite profiles of non-estrus yaks and GnRH- and PG-induced estrus in yaks, respectively. A total of 70 significantly different metabolites were detected in the non-estrus (IO) and estrus (GnRH) groups, while 77 significantly different metabolites were detected in the non-estrus (IO) and estrus (PGF) groups. GnRH and PGF respectively induced estrus in yaks and screened common metabolic pathways in the control group: biosynthesis of primary bile acids, cholesterol metabolism, and the biosynthesis of unsaturated fatty acids.

The gestation period of yaks is 250–260 days, theoretically allowing for one birth per year. However, currently, their reproduction rate is typically once every two years or twice every three years [23]. Low reproductive rate is a significant factor limiting yak production. Therefore, in order to improve the fertility of yaks and overcome the difficulty of applying artificial intelligence to yak breeding, scholars have evaluated the effects of various hormone therapy schemes on synchronous estrus and ovulation in yaks. In the 1980s, Chinese scholars commonly used the triple hormone mixture (consisting of androgen, progesterone, and estrogen) to induce estrus in yaks. Although it could successfully induce estrus and ovulation in yaks, the conception rate remained low [24]. In addition, Zi et al. [25] discovered that the P4-PGF2-PMSG protocol can efficiently trigger estrus in postpartum lactating yaks, leading to a satisfactory pregnancy rate. Meanwhile, Chen [26] used three protocols to induce estrus in female yaks: the triple hormone (ITC), progesterone vaginal sponge combined with cloprostenol (CIDR + PG), and gonadotropin-releasing hormone combined with cloprostenol (GnRH + PG + GnRH). The results showed that the estrus rate after treatment with the ITC protocol was higher than that of the other two protocols, but the conception rate after estrus was very low. Both the CIDR + PG and GnRH + PG + GnRH protocols significantly increased the estrus rate and conception rate compared to the control group. Moreover, the estrus rate and conception rate after treatment with the GnRH + PG + GnRH protocol were higher than those with the CIDR + PG protocol, indicating that the GnRH + PG + GnRH protocol was more effective in inducing postpartum anestrus in female yaks and could be determined as the optimal induction protocol. In summary, this study used gonarelin and cloprostenol sodium to synchronize estrus in yaks.

Gonarelin and cloprostenol sodium are mainly used in animal husbandry for reproductive control, inducing estrus, controlling reproductive cycles, and performing artificial insemination by affecting the levels of reproductive hormones such as androgens, estrogen, and progesterone [27,28]. There are research reports that the corpus luteum (CL) can produce a large amount of progesterone P4, which requires a large amount of cholesterol [29,30]. Cholesterol is a prerequisite for synthesizing reproductive hormones such as estrogen and progesterone, which are also essential hormones for yak estrus [29,31]. In addition, Agbugba et al. [32] found that cholesterol was predictive for proestrus, metoestrus, and dioestrus. Therefore, cholesterol levels have a significant impact on the synthesis and secretion of reproductive hormones, which in turn affect the estrus process of animals. This suggests that in this experiment, gonarelin and cloprostenol sodium regulated the synthesis and secretion of reproductive hormones such as estrogen and progesterone through the cholesterol metabolism pathway, thereby promoting estrus in yaks.

Cholic acid and chenodeoxycholic acid constitute the predominant bile acids (BAs) in the majority of livestock species [33]. Investigations have suggested a link between BAs and reproductive processes in humans and cattle. This association underscores the potential significance of BAs in modulating reproductive outcomes [34,35]. Blaschka et al. [36] discovered that cholic acid and glycocholic acid are the predominant bile acid species within follicular fluid, blood, and urine across all stages of the estrus cycle. Cholic acid, a free bile acid, is prominently found in the bile of mammals and other vertebrates. Its synthesis occurs in the liver from cholesterol and involves conversion into primary bile acids through cytochrome P450-mediated cholesterol oxidation pathways [37]. Cholic acid is also involved in regulating the metabolism of liver cholesterol, and the level of bile acid can to some extent reflect the level of cholesterol in the body, playing an important role in the cyclic metabolism of cholesterol. Cholesterol is a precursor for the synthesis of steroid hormones in the adrenal cortex, testes, and ovaries. One study found that adding cholesterol to the diet of guinea pigs increased the synthesis of cholic acid [38]. Meanwhile, the biosynthesis and metabolism of cholic acid may affect the synthesis and metabolism of hormones in animals including reproductive hormones related to estrus [39]. This study found that the bile acid content of GnRH and NE yaks was higher than that of IO yaks, indicating that bile acid intake may affect the cholesterol levels of GnRH and PGF yaks, thereby affecting the synthesis of steroid hormones and affecting their estrus.

Polyunsaturated fatty acids are a type of unsaturated fatty acid, and supplementation with polyunsaturated fatty acids (PUFAs) has been shown to increase progesterone concentration by promoting luteal cell development and alleviating liver steroid metabolism [40,41]. Feeding cows rich in essential fatty acids may lead to the synthesis of more progesterone, as larger follicles form a larger ovarian corpus luteum [42]. Staples et al. [43] found that adding dietary fatty acids could significantly improve the rate of first insemination and conception. More and more evidence suggests that the design and delivery of adding unsaturated fatty acids to the lower intestine for absorption may target reproductive tissues and alter reproductive function and fertility [44]. Tetracosanoic acid and nervonic acid were significantly different metabolites between the GnRH vs. IO group and PGF vs. IO group. Compared with the IO group, the expression of tetracosanoic acid and nervonic acid in the serum of the GnRH and PGF group was significantly upregulated, and were involved in regulating the biosynthesis of unsaturated fatty acids. Therefore, it is speculated that tetracosanoic acid and nervonic acid unsaturated fatty acids can affect the synthesis and secretion of estrus related hormones including estrogen and progesterone in the body.

## 5. Conclusions

Based on the UHPLC-Q-TOF/MS metabolomics research method, this study analyzed the serum metabolites of gonarelin and cloprostenol sodium affecting the estrus metabolism of yak. Through the screening of common differential metabolites and bioinformatics analysis, it was found that the injection of gonarelin and cloprostenol sodium caused the disturbance of six related metabolic pathways such as the beta oxidation of very long chain fatty acids and bile acid biosynthesis, involving the metabolism of fatty acids, amino acids, and lipids in the body. The results of this study provide a reference for the study of the reproductive performance of female yaks in plateau areas.

## Figures and Tables

**Figure 1 animals-14-01399-f001:**
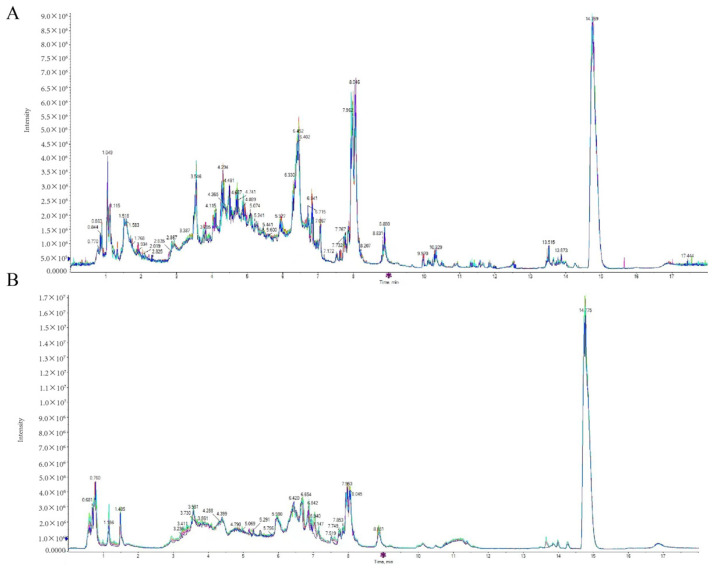
TIC overlap mapping of QC samples. (**A**) QC sample TIC overlap map in positive mode; (**B**) QC sample TIC overlap map in negative mode.

**Figure 2 animals-14-01399-f002:**
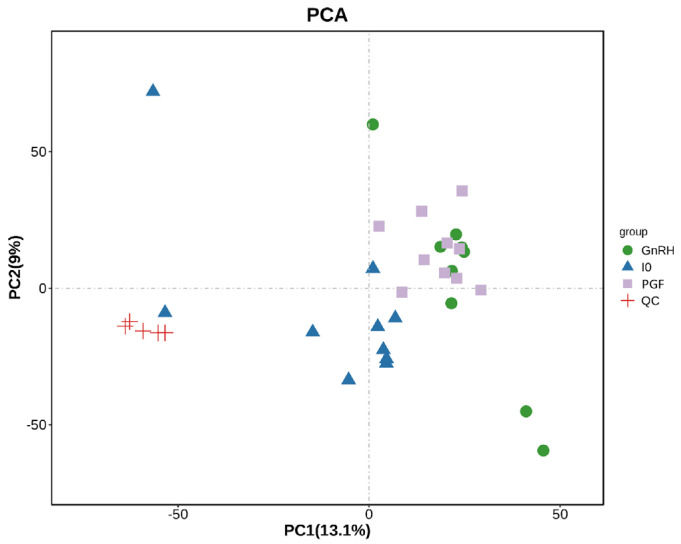
Overall sample PCA score chart.

**Figure 3 animals-14-01399-f003:**
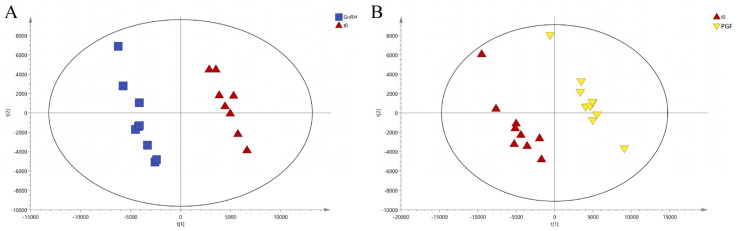
PLS-DA scores plot of all groups. (**A**) The PLS-DA score plot of GnRH vs. IO group. (**B**) The PLS-DA score plot of PGF vs. IO group.

**Figure 4 animals-14-01399-f004:**
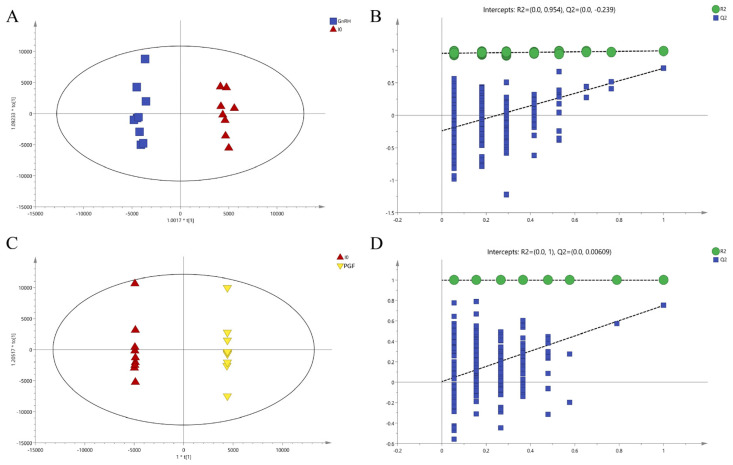
OPLS-DA scores plot and permutation test of all groups. (**A**) The OPLS-DA score plot of GnRH vs. IO group. (**B**) The permutation test for GnRH vs. IO group. (**C**) The OPLS-DA score plot of PGF vs. IO group. (**D**) The permutation test for PGF vs. IO group.

**Figure 5 animals-14-01399-f005:**
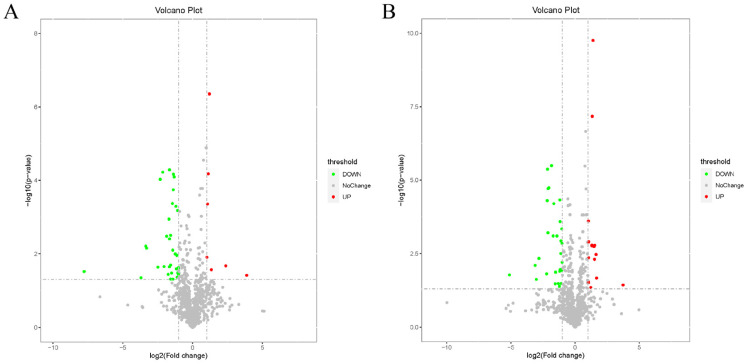
Volcano maps of differential metabolites. (**A**) Volcano plot of differential metabolites in GnRH vs. IO group. (**B**) Volcano plot of differential metabolites in PGF vs. IO group.

**Figure 6 animals-14-01399-f006:**
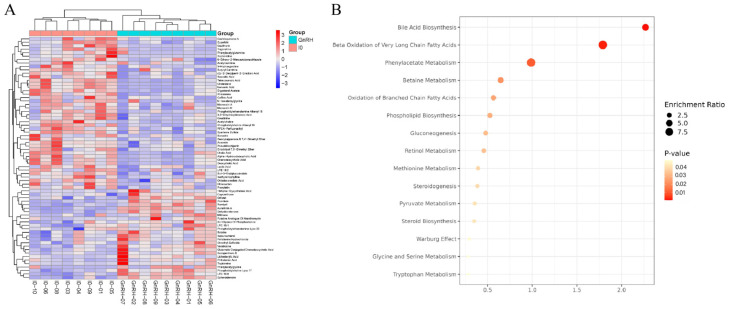
Cluster diagram and enriched pathways of significantly different metabolites between the GnRH and IO groups. (**A**) Cluster analysis of differential metabolites in GnRH vs. IO groups. (**B**) KEGG enrichment analysis of differential metabolites in the GnRH vs. IO group. The roundness color represents the *p* value. The roundness area represents the differential metabolites number in this pathway.

**Figure 7 animals-14-01399-f007:**
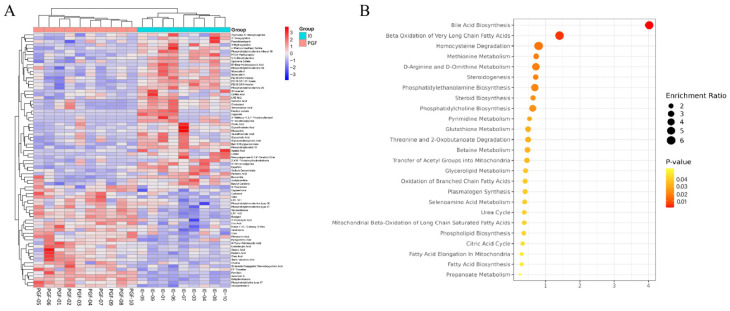
Cluster diagram and enriched pathways of significantly different metabolites between the PGF and IO groups. (**A**) Cluster analysis of differential metabolites in PGF vs. IO groups. (**B**) KEGG enrichment analysis of differential metabolites in the PGF vs. IO group. The roundness color represents the *p* value. The roundness area represents the differential metabolites number in this pathway.

**Figure 8 animals-14-01399-f008:**
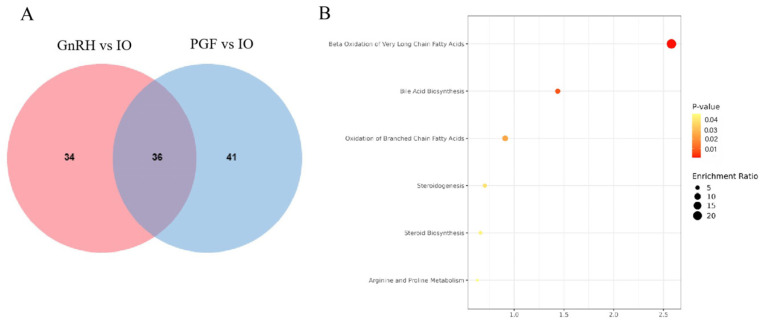
Venn distribution map and enrichment analysis of significantly different metabolites between the GnRH vs. IO group and the PGF vs. IO group. (**A**) Venn diagram of differential metabolites between GnRH vs. IO group and PGF vs. IO group. (**B**) KEGG enrichment analysis of common differential metabolism between GnRH vs. IO group and PGF vs. IO group. The roundness color represents the *p* value. The roundness area represents the differential metabolites number in this pathway.

## Data Availability

Data will be made available on request.

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
