# Peer review of "Serum Metabolomic Analysis of Synchronous Estrus in Yaks Based on UPLC-Q-TOF MS Technology"

_animals, 2024, doi:10.3390/ani14101399_

Round 1

Reviewer 1 Report

Comments and Suggestions for Authors

Dear authors,

Your article addresses an important and timely topic. The biotechnology of estrus synchronization has been used many years in cattle breeding. However, there are unclear questions related to the individual response of animals to the hormonal treating.   I found the subject matter of the article   interesting. Using modern methods of analyses and robust statistic, you obtained  interesting data about metabolites profile in serum of animals with synchronized estrus.   Really, maybe individual metabolomics response to the hormonal treatment can explain success or nonsuccess of  different protocols for the estrus synchronization.

Please, pay attention to some remarks that could help you to improve the manuscript:

- improve the English for clearly understanding of your results explanation

- in M&M explain in detail - how was the estrus detected? How many animals became in heat in each group? why did you take only 10 blood  probes among 15 animals?  

- fig.5 -    you have two parts on this figure, however there is no explanation what we see on A and B parts;

-some printing errors : Line -301 Gonarilin -"Gonarelin"

line 59-60 - " Research has reported that GnRH and PGF2α The combined use of im proved efficiency and accuracy of synchronized estrus in beef cattle "- not clear sentence with this "the"in the middle;

line 60- " GnRH 60 and PGF2α paly important roles"- maybe "play"?

line 138 - "" XCMS module within the MSDIAL"  and line 141 - HMDB - explain the abbreviation

In addition, it could be interesting to compare the metabolites profile of treated animals manifested and no manifested the estrus.  That can give you the information about potential markers of successful estrus synchronization.

Comments on the Quality of English Language

Dear authors,

For clearly understanding of your results the English of the article must be improved.

Author Response

Dear authors,

Your article addresses an important and timely topic. The biotechnology of estrus synchronization has been used many years in cattle breeding. However, there are unclear questions related to the individual response of animals to the hormonal treating. I found the subject matter of the article interesting. Using modern methods of analyses and robust statistic, you obtained interesting data about metabolites profile in serum of animals with synchronized estrus. Really, maybe individual metabolomics response to the hormonal treatment can explain success or nonsuccess of  different protocols for the estrus synchronization.

Please, pay attention to some remarks that could help you to improve the manuscript:

Reply: Thanks for your comments. We have revised the manuscript point to point according to your comments and suggestions.

- improve the English for clearly understanding of your results explanation

Reply: Thanks for your comment. We have carefully read the whole text and corrected the grammar and polished the sentences with the help of experts who are proficient in English.

- in M&M explain in detail - how was the estrus detected? How many animals became in heat in each group? why did you take only 10 blood probes among 15 animals?

Reply: Thanks for your comments. We have expalined them in Lines 122-129 according to your suggestion.

- fig.5 -you have two parts on this figure, however there is no explanation what we see on A and B parts;

Reply: Thanks for your comments. Based on your suggestions, we have explained it in Lines 259-263.

-some printing errors : Line -301 Gonarilin -"Gonarelin"

Reply: Thanks for your comments. According to your comments, we have revised it in Line 398.

line 59-60 - "Research has reported that GnRH and PGF2α The combined use of im proved efficiency and accuracy of synchronized estrus in beef cattle "- not clear sentence with this "the"in the middle;

Reply: Thanks for your comments. We have rewritten it in Lines 75-77 according to your suggestion.

line 60- "GnRH and PGF2α paly important roles"- maybe "play"?

Reply: Thanks for your comments. We have revised it in Line 77 according to your suggestion.

line 138 - "" XCMS module within the MSDIAL" and line 141 -HMDB - explain the abbreviation

Reply: Thanks for your comments. We have explained them in Line 180 and Line 184 according to your comments. In addition, XCMS is one of the most used software for liquid chromatography–mass spectrometry (LC-MS) data processing and it exists both as an R package and as a cloud-based platform known as XCMS Online.

In addition, it could be interesting to compare the metabolites profile of treated animals manifested and no manifested the estrus. That can give you the information about potential markers of successful estrus synchronization.

Comments on the Quality of English Language

Dear authors,

For clearly understanding of your results the English of the article must be improved.

Reply: Thanks for your comment. We have carefully read the whole text and corrected the grammar and polished the sentences with the help of experts who are proficient in English.

Reviewer 2 Report

Comments and Suggestions for Authors

The manuscript “Serum metabolomic analysis of synchronous estrus in yaks based on UPLC-Q-TOF MS technology” aimed to compare serum metabolites in yaks with synchronized estrus and inactive ovaries. The authors justify the importance of the work considering the low reproductive efficiency of the species and the need to know these metabolites as a way to improve the efficiency of synchronization protocols. The data is interesting and the methodology appropriately used. Specific comments are presented below:

Abstract:

1. In “Blood samples of yaks were collected from jugular vein”, inform the collection period for these samples.

2. It is necessary to insert a "summary" item, which is Animals' standard.

3. Define “PG”.

Introduction, material and methods:

1. In “Research has reported that GnRH and PGF2α The combined use of improved efficiency and accuracy of synchronized estrus in beef cattle”, re-write, as the sentence seems confusing to me.

2. Enter the period for collecting blood samples.

3. Present the centrifuge rotation values in "g" units.

Results, and discussion:

1. The images need to be improved in quality.

2. Indicate whether there are other methods that could be used to synchronize the species' estrus.

Author Response

The manuscript “Serum metabolomic analysis of synchronous estrus in yaks based on UPLC-Q-TOF MS technology” aimed to compare serum metabolites in yaks with synchronized estrus and inactive ovaries. The authors justify the importance of the work considering the low reproductive efficiency of the species and the need to know these metabolites as a way to improve the efficiency of synchronization protocols. The data is interesting and the methodology appropriately used. Specific comments are presented below:

Abstract:

  1. In “Blood samples of yaks were collected from jugular vein”, inform the collection period for these samples.

Reply: Thanks for your suggestion. We have added it in Lines 32-34 according to your suggestion.

  1. It is necessary to insert a "summary" item, which is Animals' standard.

Reply: Thanks for your comments. Based on your suggestion, we have added it in Lines 16-24.

  1. Define “PG”.

Reply: Thanks for your suggestion. We have defined it in Line 44 according to your suggestion.

Introduction, material and methods:

  1. In “Research has reported that GnRH and PGF2α The combined use of improved efficiency and accuracy of synchronized estrus in beef cattle”, re-write, as the sentence seems confusing to me.

Reply: Thanks for your comments. We have rewritten it in Lines 75-77 according to your suggestion.

  1. Enter the period for collecting blood samples.

Reply: Thanks for your suggestion. We have added it in Lines 126-129 according to your suggestion.

  1. Present the centrifuge rotation values in "g" units.

Reply: Thanks for your comments. We have expressed the centrifuge rotation value in units of "g", which showed in Line 129.

Results, and discussion:

  1. The images need to be improved in quality.

Reply: Thanks for your suggestion. Based on your comments, we have improved the quality of the images.

  1. Indicate whether there are other methods that could be used to synchronize the species' estrus.

Reply: Thanks for your comments. We have added it in Lines 355-387.

Reviewer 3 Report

Comments and Suggestions for Authors

This manuscript explores how plasma components change when yaks are induced into estrus using pharmaceutical agents. the paper is well written, although it is not clear when the samples are being taken relative to estrus.  The data are presented nicely, and a thorough discussion is provided. 

The solid experimental procedures is a definite positive, but an obvious negative for this work is that the study does not seem to be properly controlled. Yaks not in estrus were used for comparisons with the pharmaceutical agents. This seems like comparing apples with oranges.  It also brings into question what the actual objective of the work is.  Inducing estrus altered the metabolic profile. It is not clear why this important to define. it seems obvious. Authors should clarify how the experimental design fits with their hypothesis and objectives.

Minor points.

Include the month(s) of the project. Time of year seem important given that yaks are seasonal breeders.

It curious that you are abbreviating the chloroprostol group as "NE". Its not clear what that stands for.  Seems like abbreviating it "PGF" would be more fitting.

Author Response

This manuscript explores how plasma components change when yaks are induced into estrus using pharmaceutical agents. the paper is well written, although it is not clear when the samples are being taken relative to estrus. The data are presented nicely, and a thorough discussion is provided.

The solid experimental procedures is a definite positive, but an obvious negative for this work is that the study does not seem to be properly controlled. Yaks not in estrus were used for comparisons with the pharmaceutical agents. This seems like comparing apples with oranges. It also brings into question what the actual objective of the work is. Inducing estrus altered the metabolic profile. It is not clear why this important to define. it seems obvious. Authors should clarify how the experimental design fits with their hypothesis and objectives.

Reply: Thanks for your comments. According to your suggestion, We hereby provide a detailed explanation and clarification. In this study, non-estrus yaks were chosen as the control group to establish a baseline state, allowing for a more thorough observation of the changes in metabolites during hormonally induced synchronized estrus. There exist distinct physiological differences between non-estrus yaks and those undergoing hormonally induced synchronized estrus. Through metabolomics approaches, we aim to identify signature metabolites that indicate estrus in yaks.

Minor points.

Include the month(s) of the project. Time of year seem important given that yaks are seasonal breeders.

Reply: Thanks for your comments. We have stated it in Line 114 according to your suggestion.

It curious that you are abbreviating the chloroprostol group as "NE". Its not clear what that stands for. Seems like abbreviating it "PGF" would be more fitting.

Reply: Thanks for your suggestions. Based on your comments, we have revised them in the whole text.

Round 2

Reviewer 3 Report

Comments and Suggestions for Authors

all of my concerns have been addressed.